# SurvivalPath:A R package for conducting personalized survival path mapping based on time-series survival data

**Lujun Shen**[1,2☯], **Jinqing Mo**[3☯], **Changsheng Yang**[4☯], **Yiquan Jiang**[1,2☯], **Liangru Ke**[2,5], **Dan Hou**[6], **Jingdong Yan**[7], **Tao Zhang**[7]*, **Weijun Fan**[1,2]*

**1** Department of Minimally Invasive Interventional Therapy, Sun Yat-sen University Cancer Center, Guangzhou, People's Republic of China, **2** State Key Laboratory of Oncology in South China, Collaborative Innovation Center of Cancer Medicine, Sun Yat-sen University, Guangzhou, People's Republic of China, **3** Zhongshan School of Medicine, Sun Yat-sen University Cancer Center, Guangzhou, People's Republic of China, **4** Department of Spine Surgery, Third Affiliated Hospital of Southern Medical University, Guangzhou, People's Republic of China, **5** Department of Radiology, Sun Yat-sen University Cancer Center, Guangzhou, People's Republic of China, **6** Deepaint Intelligence Technology Co., Ltd., Guangzhou, People's Republic of China, **7** Department of Information, Nanfang Hospital, Southern Medical University, Guangzhou, People's Republic of China

☯ These authors contributed equally to this work.
* nyy1920@163.com (TZ); fanwj@sysucc.org.cn (WF)

**Data Availability Statement:** The SurvivalPath R package is freely available from github, https://github.com/zhangt369/SurvivalPath. The time-series dataset of patients with hepatocellular

## Abstract

### Summary

The survival path mapping approach has been proposed for dynamic prognostication of cancer patients using time-series survival data. The SurvivalPath R package was developed to facilitate building personalized survival path models. The package contains functions to convert time-series data into time-slices data by fixed interval based on time information of input medical records. After the pre-processing of data, under a user-defined parameters on covariates, significance level, minimum bifurcation sample size and number of time slices for analysis, survival paths can be computed using the main function, which can be visualized as a tree diagram, with important parameters annotated. The package also includes function for analyzing the connections between exposure/treatment and node transitions, and function for screening patient subgroup with specific features, which can be used for further exploration analysis. In this study, we demonstrate the application of this package in a large dataset of patients with hepatocellular carcinoma, which is embedded in the package. The SurvivalPath R package is freely available from CRAN, with source code and documentation hosted at https://github.com/zhangt369/SurvivalPath.

### Author summary

Patients with advanced stage malignancies always require repeated disease surveillance and treatment. However, in the era of big data, we still lack non-black-box methodology to utilize the time-series clinical data to actualize dynamic survival prediction and

carcinoma for demonstration is embedded in the package.

**Funding:** This work was funded by Talent Cultivation Project (sponsored by Guangdong Medical Equipment Society Medical Informatization Yangcheng Forum and Xinhua Three Technology Co., Ltd., project number GDYZ2021A05 to TZ); and Sun Yat-sen University Youth Development Project (2019): Development of R package of survival path mapping and its implementation in personalized treatment of HCC (No. 19ykpy200 to LS). The funders had no role in study design, data collection and analysis, decision to publish, or preparation of the manuscript.

**Competing interests:** The authors declare no competing interest.

treatment planning. In our previous study, our team introduced a new analytical approach called "Survival Path" to interpret the time-series data of cancer patients. The paths constructed for specific malignancy could serve as a "map" for dynamic prognosis prediction and treatment planning. Here we introduced the SurvivalPath R package, which aims to promote the standardization of this methodology and allow users to build personalized survival path models using time-series survival data. This tool is expected to facilitate dynamic clinical decision for oncologists, as well as to promote the rationale change from "Guideline-based Medicine" to "Data-driven Medicine".

This is a *PLOS Computational Biology* Software paper.

## 1. Introduction

The idea of developing the SurvivalPath R package stems from our previous exploratory work, in which we attempted to achieve dynamic prognosis prediction by establishing survival paths based on the time-series data of patients with hepatocellular carcinoma (HCC) [1]. In the last decade, enormous efforts have been made in establishing prognostic models using machine learning or deep learning techniques with large dataset to facilitate dynamic management of cancer patients [2][3]. However, most of these models are black-box, which limit their interpretability and clinical application [4][5]. The survival path approach we proposed provide a potential solution for dynamic prognosis prediction and management of cancer patients by constructing survival path maps using returned key prognostic factors after analysis of structured time-series survival data. More importantly, the survival path model could be easily understood and utilized by clinicians when compared to black-box models.

The SurvivalPath R package is a newly developed tool to facilitate fast building of survival path models, with an aim of promoting standardization of this methodology. To make the SurvivalPath R package adapted to datasets in whom the included variables have potential collinearity [6], we optimized the feature selection process. One-to one collinearity analysis was embedded to screen out noncollinear candidate variables before formal feature selection in the main function. This step simplifies the process of picking variables among different categorical variables and reduces the confounding impact of potential collinearity on feature selection in the Cox model. In addition, the SurvivalPath R package is compatible with continuous variable, enabling automatic binary classification of continuous variables and their entry into the model [7]. To facilitate fast building and efficient usage of survival path, this package also provide functions to enable: 1. pre-processing of the data into regular time slices; 2. fast selection of patient subgroups based on personalized design; 3. demonstration of the connections between exposure/treatment and node transitions.

## 2. Design and implementation

### Ethics statement

This study was approved by Sun Yat-sen University Cancer Center's (SYSUCC) Ethics Committee (no. B2022-220-01). The usage example presented was an anonymously retrospective analysis of routine data and therefore we requested and were granted a waiver of individual informed consent from the SYSUCC ethics committee.

## Software design

**1) Converting data of time-slices into data of time slices.** The time series data of each patient firstly needs to be sorted out in rows with each row represent data at specific time point. The time axis needs to be divided into time slices with the proposed interval given by the researcher. For each patient, the zero point was set at first/earliest time point, which is usually the first row of the patient's time-series data. The time slice with complete data of all included variables was defined as slice with complete data (Fig 1A). This process of data pre-processing could be finished using the built-in function "timedivision" in this package.

**2) Feature selection and survival path mapping.** The key features for bifurcation for each node in the construction of the survival paths are selected by repeated cycles of processing flow. The processing flow in each node includes the following steps.

Step 1: Calling the data in the parent node. Initially, the data of all included cohort at no.1 time slice was called, which is denoted $S_{(all;\ ts\ =\ 1)}$ (Fig 1B). Univariate analysis with Kaplan–Meier (KM) method was utilized to identify candidate variables ($X_1$, $X_2$, $X_a$, $X_b$,..., $X_p$). In constructing the survival path, to control the false discovery rate and create a balanced survival

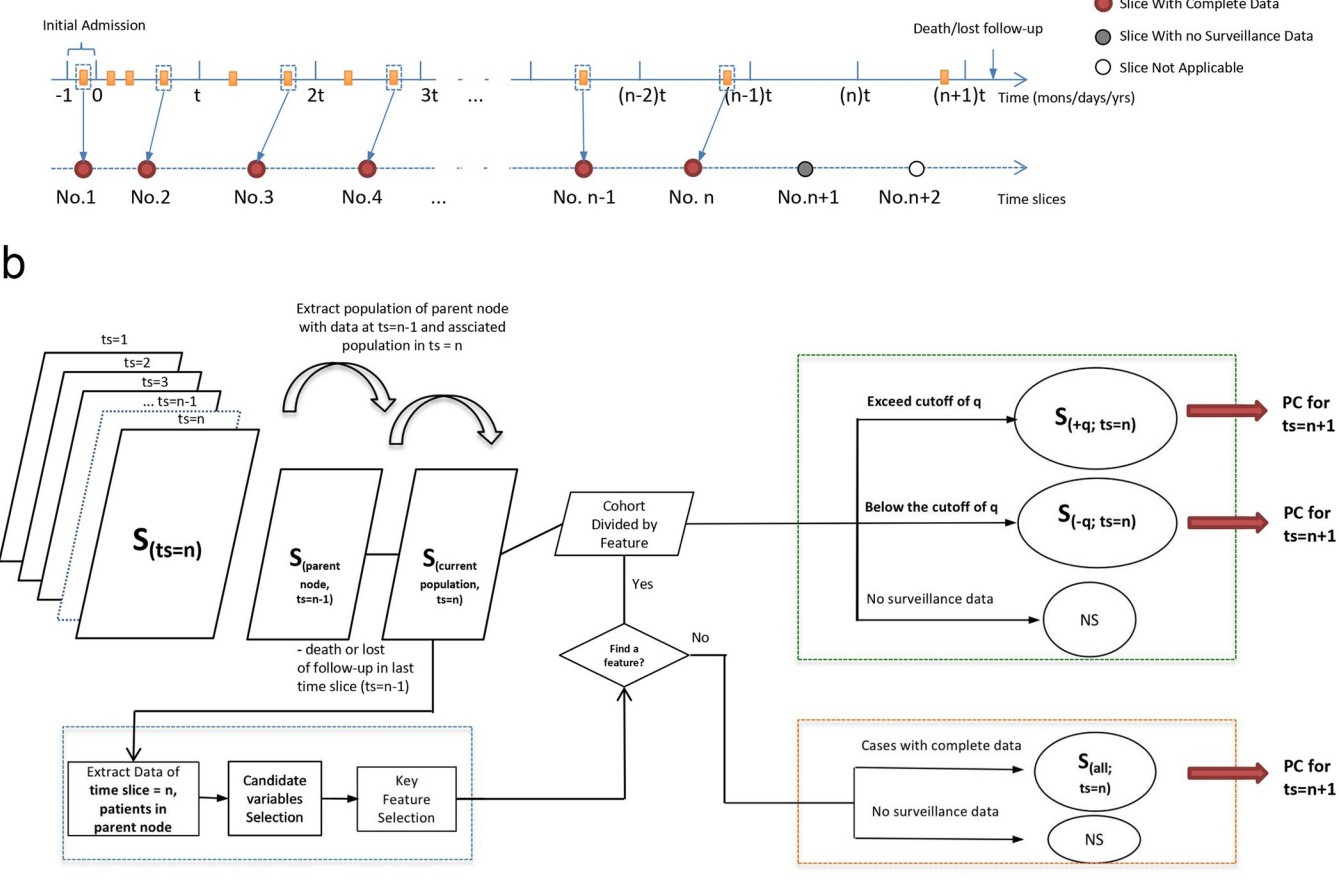

**Fig 1.** Flowchart of analysis for survival path mapping. (A) The time-series data of participants were converted into data of time slices with constant time interval. (B) Starting from the first time slice, data of the whole population would initially undergo the processing cycle (PC) for feature selection and subgroup subdivision, followed by PC in each subdivided subgroup data at the next time slice. The subdivision of parent node in the No. (n-1) time slice determine the population of the nodes in No.n time slice. The program continue until user defined last time slice.

path, the researcher can set the significance level for feature selection in the argument of p. value in the main function survivalpath().

Step 2: Judgment. The minimum sample size required for the Cox proportional hazard regression model with multiple covariates was input with the parameter "Minsample" in the main function Survivalpath ().

Step 3: Feature selection. All the significant variables detected using the KM method will undergo collinearity analysis between pairs. The correlation coefficient between variables in pairs was calculated by the formula below, where $X_a$ and $X_b$ represent one of the variables, respectively.

$$r = \frac{\sum_{i=1}^{n} (x_{ai} - \bar{x}_a)(x_{bi} - \bar{x}_b)}{\sqrt{\sum_{i=1}^{n} (x_{ai} - \bar{x}_a)^2}\sqrt{\sum_{i=1}^{n} (x_{bi} - \bar{x}_b)^2}} \tag{1}$$

The researcher can set the correlation coefficient by himself, and the pair of variables that exceed the correlation coefficient will automatically compare their Akaike information criterion (AIC) values when each of two serve as the only predictor for outcome; the variable with the smaller AIC value will be removed. All the significant variables left were the "candidate variables" and then be put into Cox proportional hazard regression model, which assumes the hazard as follows,

$$h(t) = h_0(t)\exp\left(\sum_{j=1}^{p} \beta_j x_j\right) \tag{2}$$

where $(X_1, X_2, \ldots, X_p)$ is a vector of p predictor variables, and $\beta_1, \beta_2, \ldots, \beta_p$ are the corresponding regression coefficients, which are the weights given to each variable by the model. The original model included all the candidate variables and was presented as follows:

$$Y = \beta_0 + \beta_1 X_1 + \ldots + \beta_{r-1} X_{r-1} + \varepsilon \tag{3}$$

The backward elimination (BE) procedure was carried out, with the following p tests, $H_{0j}$: $\beta_j = 0$; j = 1; 2,. . ., p, the lowest partial F-test value $F_l$ corresponding to $H_{0l}$: $\beta_l = 0$ is compared with the pre-set significance values $F_0$. If $F_l < F_0$, then $F_l$ is deleted and the new original model is:

$$Y = \beta_0 + \beta_1 X_1 + \ldots \beta_{l-1} X_{l-1} + \beta_{l+1} X_{l+1} + \beta_{r-1} X_{r-1} + \varepsilon \tag{4}$$

Then, a stepwise BE procedure was continued, until all $F_l > F_0$, and the model is what we choose. The importance of each variable in the fixed Cox model can be obtained as follows:

$$\Gamma_q = -2\log(L_h/L_{h-q}) \tag{5}$$

where $L_h$ refers to the likelihood of the fixed model and $L_{h-q}$ refers to the likelihood of model after elimination of the variable $X_q$. The variable eliminated from the model with the maximal change of $-2\log(L)$ was selected.

Step 4: Cycles of processing. Based on the selected dichotomized variable $X_q$, the cohort $S_{(all; ts = 1)}$ will be divided into two subgroups $S_{(-q; ts = 1)}$ and $S_{(+q; ts = 1)}$. The two subgroups serve as parent node and the corresponding cohort at the next time slice (after removal of those lose follow-up or reach endpoint in last time slice) (Fig 1B). If no variable is selected in certain node, the cohort in that node stays the current classification and the data in next time slice will repeat step 1–3. The process flow will end until last time slice, which could be assigned with the argument "time_slices" in the main function.

Step 5: Graphic representation: A tree-based survival path can be constructed.

**3) Dealing with continuous variable: Automatic dichotomization.** A large number of studies have shown that the method based on time-dependent receiver operating characteristic (ROC) curves can accurately find the ideal cutoff value of continuous variables [8]. Therefore, we embed the function for automatic dichotomization of the continuous variables according to the best cutoff recommended by the Time-dependent ROC curve in the **classifydata ()** function, thus the program is compatible with candidate features in the form of continuous variables.

**4) Survival Path Mapping of Re-arranged data.** In the past, researchers often used the traditional staging system for dynamic staging, that is, patients with similar disease states at different time points after the onset of disease are considered to have disease with similar severity [9]. Based on the concept that similar disease status at different time points across patients can sometimes be regarded as similar diseases and set as the same starting line, we designed the **matchsubgroup** () function, which can capture the time-series data of patient once the disease status happen from the time series data through matching values of key features.

## Implementation

Before constructing a survival path model, time-series dataset in the form of continuous time slices is needed. The prepared time-series data should meet the following requirements:

1. The time series data of each participant is arranged in rows with each row represent data at each specific time point.

2. The dataset need to contain separate fields for participants' ID, time points of the recorded data, survival time and survival outcome. The records in each row need to have complete data on these fields.

3. Dichotomies as candidate variables are recommended. When the input variables are grade variables or continuous variables, re-classification is feasible by setting argument "*ifclassifydata = TRUE*" in the function generatorDTSD(). Multiple categorical variables with more than three categories that are not logically correlated or in rank correlation are not recommended as the interpretation of result will be difficult.

To build a survival path model using SurvivalPath R package, four key functions are important in implementation. The first and the main function is survivalpath(), whose function signature is *survivalpath (DTSD, time_slices, treatments = NULL, num_categories = 2, p. value = 0.05, minsample = 15, degreeofcorrelation = 0.7, rates = 365)*. The DTSD is a list object that can be generated using the built-in function *generatorDTSD()*, which includes list data of *time*, *status*, *timeslicedata* and *tspatientid* of each time slice. The *time_slices* argument specify the total number of time slices (starting from the front) need to be included in the survival path model. The default value of *treatments* is NULL, this argument can specify the intervention/exposure variable the researcher find interest, which could be utilized during analysis of efficacy of different arms in certain nodes. The last five arguments were used to specify the maximum number of branches that each node can divide, p.value for hypothesis testing, minimum sample size for branching, cutoff value of correlation coefficient for collinearity analysis and the time point set for estimating survival rates of each node, respectively.

The second function timedivision() is used for data pre-precessing, whose signature is *timedivision (dataset, ID, time, period = 30, left_interval = 0.5, right_interval = 0.5)*.

The first argument *dataset* specify the dataframe of row arranged time-series survival data, which is usually un-preprocessed data prepared by the researcher; the data structure

**Fig 2.** The data structure needed to construct survival path model. The columns includes time points, observational variables, treatment variables (optional) and outcome variables. Each row represent observation data at specific time point, and the date is sorted in ascending order for each participant/patient.

requirement is displayed in Fig 2. The *ID* and *time* arguments correspond to the specified variables that represent the unique identifier of each participant in each row and the date to which each data point belongs, respectively. The last three arguments were parameters used to specify the length of time slices and the time interval for inclusion of time point data into each time slice.

The third function generatorDTSD() is used to generate DTSD class objects using a preprocessed dataframe arranged in time slices. The function signature of generatorDTSD() is

generatorDTSD(dataset, periodindex, IDindex, timeindex, statusindex, variable, ifclassifydata = TRUE, predict.time = 365, isfill = TRUE)

The first argument *dataset* refers to the dataframe of time-series observations, containing identification numbers of each subject, index of time slice, value of risk factors, survival time, and survival outcomes; the *dataset* usually could be returned by timedivision() function. The arguments *periodindex*, *IDindex*, *timeindex*, *statusindex* specify the time slice indicator, ID indicator, time and status indicators in the *dataset*, respectively. The argument *variable* is a list object to specify the risk factors required for modeling. Arguments *ifclassifydata*, *predict. time* are optional and useful in automatic dichotomizing risk factors. The argument *isfill* is a logical value, which is used to confirm whether to fill in missing data. If it is True, then fill with the average level.

The last key function matchsubgroup () is used for screening and extracting data of subjects that meet the given conditions, whose function signature is

matchsubgroup(DTSD, varname, varvalue)

The first argument DTSD specifies a list object that can be generated using generatorDTSD (). The argument *varname* specifies a of list variables used to screen subjects, and the variables needs to be included in the DTSD object. The *varvalue* is defined as a list object, and subjects whose *varname* variables are equal to the values of *varvalue* will be selected. This function could be utilized for generating personalized survival path model for participant with certain features.

## 4. Results

In this section, we demonstrate the usage of SurvivalPath package through analyzing a dataset of patients with hepatocellular carcinoma (HCC).

### 4.1 Pre-processing of time-series data

The pre-processing of the time-series data could be accomplished manually or using the built-in functions. Regard the built-in function approach, first, by setting the parameters on periods

of time slices, left interval and right interval for data inclusion, we can convert the original dataset of "time points" into dataset of "time slices".

*R> data("DTSDHCC") #dataset of time points*

*R> dataset = timedivision(DTSDHCC,"ID","Date",period = 90,left_interval = 0.5, right_interval = 0.5)*

The new dataset returned in the form of time slices with one columns added: "time_slice". Usually, the number of rows in the converted dataset will be less than the original dataset after screening out records of ineligible time points. After finishing converting the dataset, we can check the amount of data at each time slice.

*R> table(dataset$time_slice)*

*1 2 3 4 5 6 7 8 9 10 11 12 13 14 15*

*2360 1586 976 627 421 284 181 119 65 24 10 2 2 2 1*

Here we found the sample size in each time slice gradually decreased. To ensure the sample size of analysis, the survival paths we constructed will focus on No.1 to No.8 time slices. Then, we extract the data of key variables of each time slice and compile them into an object of class Dynamic Time Series Data (DTSD), which is designed to run survival path mapping using generatorDTSD() function. The key variables includes survival outcomes, the variables of interest, the ID of corresponding participant of each time slice.

*R> DTSDdata <-generatorDTSD(dataset, periodindex = "time_slice", IDindex = "ID", time-index = "OStime_day", statusindex = "Status_of_death", variable = c ("Age", "Amount.of. Hepatic.Lesions", "Largest.Diameter.of.Hepatic.Lesions",*

*"New.Lesion","Vascular.Invasion", "Local.Lymph.Node.Metastasis", "Distant.Metastasis", "Child_pugh_score", "AFP"), ifclassifydata = TRUE, predict.time = 365\*1)*

As several variables included in building survival paths were continuous variable, here we use a build-in argument "ifclassifydata = TRUE" to convert these variables into dichotomies, which facilitates automatic identification of cutoff through time-dependent ROC curve estimation. To run this function, *survivalROC* package is required.

The function returns a DTSD class object for conducting survivalpath() function. The object includes 4 list objects (time, status, tsdata, tsid), which correspond to the associated data of each time slice. The object have two parameters (length and ts_size) in describing the number of time slices and sample size of each time slice. The list "cutoff" stores the calculated cut-offs of continuous variables that have been dichotomized.

## 4.2 Construct a survival path model using main function

After finishing the preparation of data, we can input the data into the main function, namely the *survivalpath()*. The investigator need to set the number of time slices when constructing the survival paths. The investigator could set p.value for feature selection process, the default value is 0.05. The investigator could also set the value of correlation coefficient to screen out noncollinear candidate variables, whose default value is 0.7. As we had already prepared this variable, we included it in our demonstration case. To view 1year OS rates on the graph, we set the *rates* of 365.

*R> result <- survivalpath(DTSDdata,time_slices = 8, p.value = 0.01, degreeofcorrelation = 0.5)*

The main function will return a dataset that indicate the steps each participant take at each time slice, a time-slices dataset with node annotation and a treedata that could be used for visualization the survival paths.

*R> View(result$data) #dataset that indicate the steps each participant take*

*R> View(result$df) #dataset with node annotation*

*Using the function* ggtree (), *we can draw a two-dimentional graph with the information of node name, sample size, median survival time, and survival rates as selected previously annotated.*

*R>library(ggplot2)*

*R>library(ggtree)*

*R> mytree <- result$tree*

*R> ggtree(mytree, color = "black",linetype = 1,size = 1.2,ladderize = T,) +*

*theme_tree2() +*

*geom_text2(aes(label = label),hjust = 0.6, vjust = -0.6, size = 3.0)+*

*geom_text2(aes(label = paste(node,size,mytree@data$survival,mytree@data$survivalrate,sep = "/")), hjust = 0.6, vjust = -1.85, size = 3.0)+*

geom_point2(aes(shape = isTip, color = isTip), size = mytree@data$size%/%200+1,show. *legend = F)+*

*labs(size = "Nitrogen",*

> *x = "TimeSlices",*
>
> *y = "Survival",*
>
> *subtitle = "node_name/sample number/Median survival time/Survival rate",*
>
> *title = "Survival Tree") +*
>
> *theme(legend.*title = element_blank(),legend.position = c(0.1,0.9))

Here we get the graph of survival path mapping, which includes all the significant bifurcations identified (Fig 3).

## 4.3 Survival Analysis and correlation analysis based on survival path

The built-in functions *plotKM(), compareTreatmentPlans() and EvolutionAfterTreatment()* are utilized for data analysis in survival paths. Once we finish the construction of survival paths model, we can generate survival curves comparing survival of patients in any of the nodes. Here we compare the survival curves of no. 24 and no. 36 nodes as a demonstration (Fig 4A).

*R> treepoints = c(24,36)*

*R> plotKM(result$data, treepoints,mytree,risk.table = T)*

We can also draw survival curves for any of the nodes to investigate the correlation between treatment/exposure and survival (Fig 4B).

*R> treepoints = c(24)*

*R> compareTreatmentPlans(result$data, treepoints,mytree,dataset,"Resection")*

We can investigate the correlation between treatment/exposure and node transition based on the survival paths map generated.

*R> treepoint = 24*

*R> A = EvolutionAfterTreatment(result$data, treepoint, mytree, dataset,"Embolization")*

*R> mytable <- xtabs(~ `Embolization`+treepoint, data = A)*

*R> prop.table (mytable,1)*

The returned results is displayed as bellowed:

*treepoint*

*Embolization 25 31 Missing follow-up*

*0 0.4654832 0.1104536 0.4240631*

*1 0.6138996 0.1441441 0.2419562*

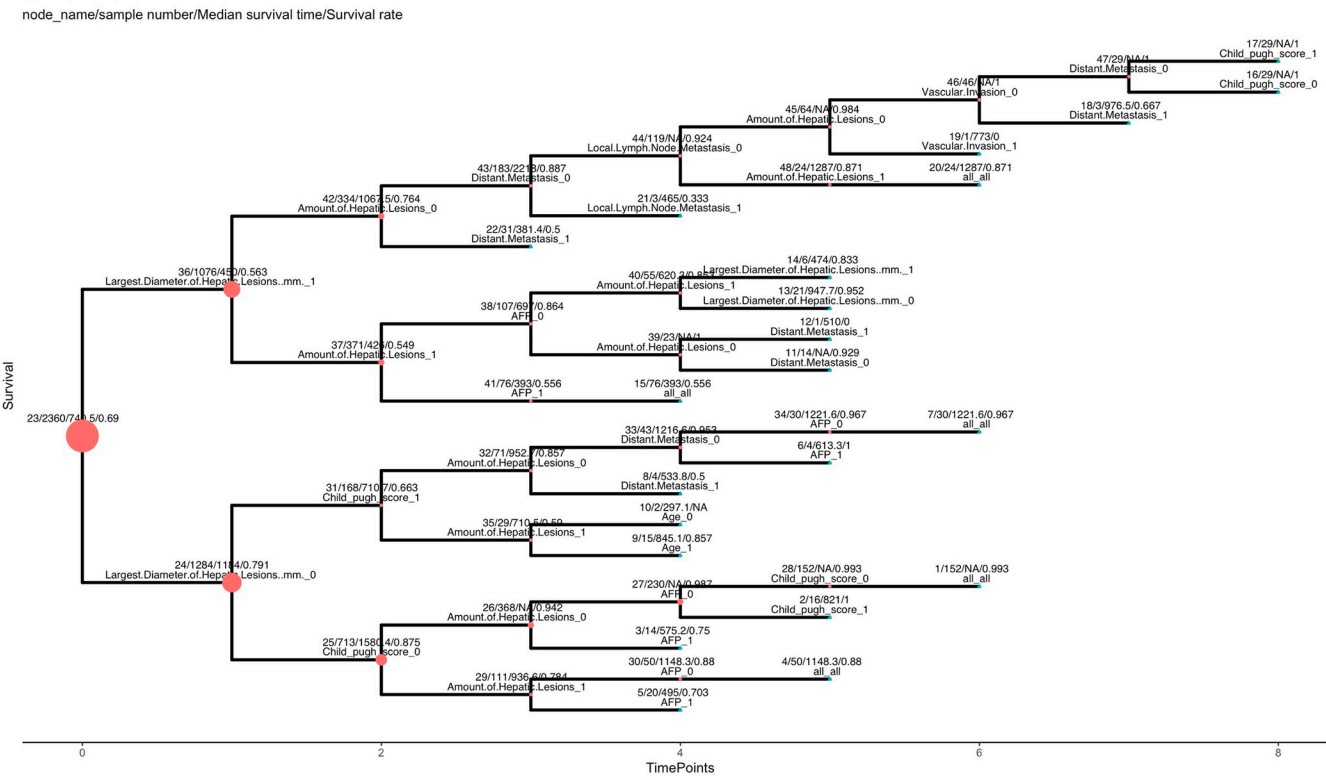

**Fig 3. Graph of survival path computed as demonstration.** The graph was built using ggtree based on tree-structure data.

The results showed that in node no.24, 46.5% patients transit to node no.25, 11.0% transit to node no.31 and 42.4% patients lost follow-up in the next time slice without embolization treatment; while 61.4% patients transit to node no.25, 14.4% transit to node no.31 and 24.4% patients lost follow-up in the next time slice after embolization treatment.

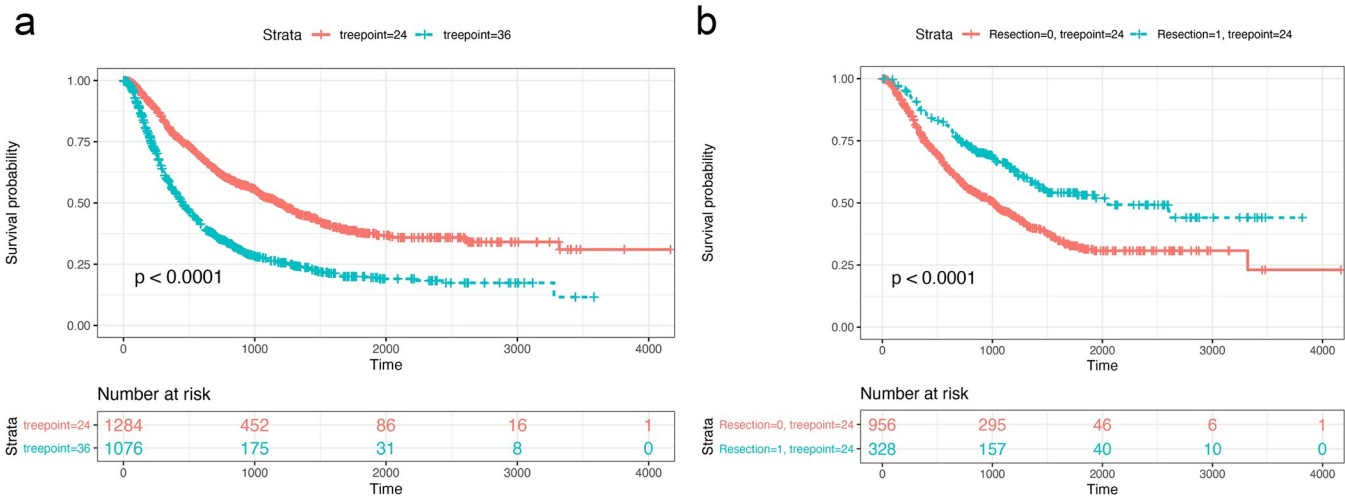

**Fig 4.** Demonstration of survival curves in comparing survival between different nodes (a) and subgroups with different treatment arms at specific nodes (b).

### 4.4 Personalized survival path mapping through features matching

For selected patients with specific disease conditions, we can extract time slices data of participants with the onset of similar conditions across time slices and construct a personalized survival path.

> *R>varname = list('Amount.of.Hepatic.Lesions',"Largest.Diameter.of.Hepatic.Lesions")*
> *R>varvalue = list(1,1)*
> *R>df <- matchsubgroup(DTSDdata,varname = varname, varvalue = varvalue)*
> *R>result<-survivalpath(df, time_slices = 4)*

### 4.5 Evaluate the survival path model with c-index

The evaluate () function was built to evaluate the discriminative ability of the survival path in each specified time slice using Harrell's concordance index C-index based on the survival path model.

> *R> evaluate(survivalpath = result, minnodesize = 15)*

The returned c-index for each time slice is displayed as bellowed:

$Cindex[[1]] 0.6008216

$Cindex[[2]] 0.6425235

$Cindex[[3]] 0.6717504

$Cindex[[4]] 0.6726773

$Cindex[[5]] 0.6725269

$Cindex[[6]] 0.6229113

### 4.6 Export data with node annotation

The time slice data with annotation of node information is stored in the results data after computation by the main function and could be exported for further usage and analysis.

> *R> write.csv(result$df, file = "output.csv", row.names = FALSE)*

## 5. Availability and future directions

In this study, we have introduced the SurvivalPath R package. The package facilitates fast generation of personalized designed survival path models, which could be used for dynamic prognosis prediction. Survival paths can be computed using the main function *survivalpath()*, under a user-defined parameters on covariates, significance level, minimum bifurcation sample size and number of time slices for analysis. The tree-structure result can be visualized as a tree diagram using ggtree R package, with important parameters annotated. We also designed built-in functions *plotKM()*, *compareTreatmentPlans()* and *EvolutionAfterTreatment()* to facilitate comparison of survival between nodes, evaluate exposure/treatment and survival, and explore the connection between exposure/treatment and node transitions, respectively. Moreover, function for picking up patient subgroup with specific features within the survival path model was created, which can be used to generate refined survival paths for further exploration analysis.

In future work, we will keep on developing the *survivalpath()* function that will facilitate node bifurcation through risk assessment that integrated multiple prognostic factors. Besides, we will keep on optimizing the visualization of survival paths through drawing 3-D graph. Lastly, we will set up cloud database for users to upload time-series data of cancer patients, which aims to be open to all users.

## Acknowledgments

We would like to thank Ms. Juan Nie for providing continuous encouragement to Dr. Lujun Shen in the past three years. We would like to thank Mr. Xueting Shen and Mrs. Meizhen Zhu for giving their best love to their son, Lujun Shen.

## Author Contributions

**Conceptualization:** Lujun Shen, Changsheng Yang, Yiquan Jiang, Tao Zhang, Weijun Fan.

**Data curation:** Lujun Shen, Jinqing Mo, Yiquan Jiang, Liangru Ke, Dan Hou, Jingdong Yan, Tao Zhang, Weijun Fan.

**Formal analysis:** Lujun Shen, Jinqing Mo, Changsheng Yang, Liangru Ke, Dan Hou, Jingdong Yan, Tao Zhang.

**Funding acquisition:** Lujun Shen, Weijun Fan.

**Investigation:** Lujun Shen, Changsheng Yang, Yiquan Jiang, Dan Hou, Jingdong Yan, Tao Zhang, Weijun Fan.

**Methodology:** Lujun Shen, Jinqing Mo, Changsheng Yang, Liangru Ke, Tao Zhang.

**Project administration:** Lujun Shen, Changsheng Yang, Yiquan Jiang, Dan Hou, Jingdong Yan, Tao Zhang, Weijun Fan.

**Resources:** Weijun Fan.

**Software:** Yiquan Jiang, Tao Zhang.

**Supervision:** Lujun Shen, Liangru Ke, Jingdong Yan, Tao Zhang, Weijun Fan.

**Validation:** Lujun Shen, Jinqing Mo, Changsheng Yang, Liangru Ke, Jingdong Yan, Tao Zhang, Weijun Fan.

**Visualization:** Lujun Shen, Changsheng Yang, Tao Zhang.

**Writing – original draft:** Lujun Shen, Jinqing Mo, Changsheng Yang, Yiquan Jiang, Liangru Ke, Jingdong Yan.

**Writing – review & editing:** Lujun Shen, Tao Zhang, Weijun Fan.

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
