## [Decision Letter · Decision Letter 0]

26 Oct 2022

Dear Prof Fan,

Thank you very much for submitting your manuscript "SurvivalPath:A R package for conducting personalized survival path mapping based on time-series survival data" for consideration at PLOS Computational Biology. As with all papers reviewed by the journal, your manuscript was reviewed by members of the editorial board and by several independent reviewers. The reviewers appreciated the attention to an important topic. Based on the reviews, we are likely to accept this manuscript for publication, providing that you modify the manuscript according to the review recommendations.

Sincerely,

Feng Fu

Academic Editor

PLOS Computational Biology

Feilim Mac Gabhann

Editor-in-Chief

PLOS Computational Biology

Reviewer's Responses to Questions

**Comments to the Authors:**

Reviewer #1: The auther develop a SurvivalPath R package to facilitate building personalized survival path models. The package contains functions to convert time-series data by into time-slices data by fixed interval based on time information of input medical records. And I have noticed that this R package is based the work which has published. However, I still have some questions:

1: How does the SurvivalPath R package deal with the missing data?

2. In Figure 2, I want to know that whether SurvivalPath R package is suitable for time-series data with a three-class outcome(eg: alive, progress, death) and one continuous variable(eg: ALBI score)

3: In Figure 3, graph of survival path is too complex for doctor to use. If it is possible, the number of survival path should be less than 5.

Reviewer #2: The authors submit an interesting manuscript (PCOMPBIOL-D-22-01103) about conducting personalized survival path mapping based on time-series survival data for possible publication in PLOS Computational Biology.

The authors introduced the SurvivalPath R package for dynamic prognostication of cancer patients using time-series survival data. I have installed the SurvivalPath R package, and verified that it works to facilitate building personalized survival path models. Congratulations to them!

I believe that readers could benefit from this package which contains functions to convert time-series data by into time-slices data by fixed interval based on time information of input medical records. More importantly, under a user-defined parameters on covariates, significance level, minimum bifurcation sample size and number of time slices for analysis, survival paths can be computed using the main function, which can be visualized as a tree diagram, with important parameters annotated. Besides, the package can also be used for further exploration analysis, which I think is an important application of this methodology.

One concern is how this package deal with the situations that specific variable with multiple important cutoffs. I notice that the optimal cutoff of continuous variable could be automatically computed, while for different situations and in different time slices, the optimal cutoff could be different.

Generally, the manuscript is well organized and easy to follow. I think it can be accepted after language polishing.

**Have the authors made all data and (if applicable) computational code underlying the findings in their manuscript fully available?**

Reviewer #1: Yes

Reviewer #2: Yes

PLOS authors have the option to publish the peer review history of their article (what does this mean?). If published, this will include your full peer review and any attached files.

Reviewer #1: No

Reviewer #2: No

Figure Files:

Data Requirements:

Reproducibility:

References:

---

## [Decision Letter · Decision Letter 1]

21 Dec 2022

Dear Prof Fan,

We are pleased to inform you that your manuscript 'SurvivalPath: A R package for conducting personalized survival path mapping based on time-series survival data' has been provisionally accepted for publication in PLOS Computational Biology.

Best regards,

Feng Fu

Academic Editor

PLOS Computational Biology

Feilim Mac Gabhann

Editor-in-Chief

PLOS Computational Biology

Reviewer's Responses to Questions

**Comments to the Authors:**

Reviewer #1: I believe this revised manuscript is suitable to accept.

**Have the authors made all data and (if applicable) computational code underlying the findings in their manuscript fully available?**

Reviewer #1: Yes

PLOS authors have the option to publish the peer review history of their article (what does this mean?). If published, this will include your full peer review and any attached files.

Reviewer #1: No

---

## [Editor Report · Acceptance letter]

29 Dec 2022

PCOMPBIOL-D-22-01103R1 

SurvivalPath: A R package for conducting personalized survival path mapping based on time-series survival data

Dear Dr Fan,

I am pleased to inform you that your manuscript has been formally accepted for publication in PLOS Computational Biology. Your manuscript is now with our production department and you will be notified of the publication date in due course.

With kind regards,

Zsofia Freund
